# Is There a Link between Thyroid Peroxidase Gene Promoter Polymorphisms and Autoimmune Thyroiditis in the Polish Population?

**DOI:** 10.3390/ijms25063312

**Published:** 2024-03-14

**Authors:** Katarzyna Lacka, Adam Maciejewski, Piotr Jarecki, Waldemar Herman, Jan K. Lacki, Ryszard Żaba, Michał J. Kowalczyk

**Affiliations:** 1Department of Endocrinology, Metabolism and Internal Medicine, Poznan University of Medical Sciences, 60-355 Poznan, Poland; 2General Practitioner Unit, 67-400 Wschowa, Poland; 3Outpatients Unit for Endocrine Diseases, 67-400 Wschowa, Poland; 4Department of Medicine, The Jacob of Paradies University, 66-400 Gorzow Wielkopolski, Poland; 5Department of Internal Medicine, Collegium Medicum, University of Zielona Gora, 65-417 Zielona Gora, Poland; 6Department of Dermatology and Venereology, Poznan University of Medical Sciences, 60-355 Poznan, Poland

**Keywords:** autoimmune thyroiditis, thyroid peroxidase, polymorphism, autoantibodies, TPOAb, autoimmunity

## Abstract

(1) Autoimmune thyroiditis (AIT) is the most common cause of primary hypothyroidism and one of the most frequent organ-specific autoimmune diseases. Its pathogenesis is polygenic and still requires further research. The aim of the study was to assess, for the first time in the Caucasian population, the role of selected *TPO* gene promoter polymorphisms (rs2071399 G/A, rs2071400C/T, rs2071402 A/G, and rs2071403 A/G) in the development of AIT. A total of 237 patients diagnosed with AIT and 130 healthy controls were genotyped for four *TPO* gene polymorphisms, and the results were statistically analyzed to check for the role of these polymorphisms. There were no significant differences in the genotype and allele frequencies of the studied *TPO* gene promoter polymorphisms between patients and controls (*p* > 0.05). The haplotype distribution (rs2071400–rs2071402–rs2071403) between the two studied groups was similar for the most common variants (CGA, CAG, TGG). Only a rare haplotype (CGG) occurred more frequently among patients compared to controls (*p* = 0.04). The studied *TPO* gene promoter polymorphisms did not show an association with susceptibility to AIT in the Caucasian Polish population, contrary to the results in Japanese patients.

## 1. Introduction

Autoimmune thyroiditis (AIT), also called Hashimoto’s disease, is the most common reason of primary hypothyroidism, and its frequency is estimated at 5–7.5%, depending on population, time when the study was conducted, diagnostic criteria, etc. [1,2]. In Poland and Europe, its incidence is estimated at 5 and 8% of the general population, respectively [2]. It affects women approximately 4–10 times more often than men [2,3]. There are also some specific population groups (e.g., women diagnosed with polycystic ovary syndrome) in which the incidence of the disease may be as high as 18–40% [4]. The prevalence of thyroid peroxidase antibodies (TPOAb) is higher compared to AIT, as they can be detected in 6.8 to 13% of the general population and up to 20% of pregnant women [5].

The diagnosis of AIT can be made on the basis of the presence of elevated serum TPOAb and/or thyroglobulin antibodies (TgAb), hypoechoic parenchyma in thyroid ultrasound, sometimes with nodules, and characteristic microscopic results in cytology. The majority of patients present with hypothyroidism, but they can also be euthyroid or, rarely, hyperthyroid. The main antithyroid antibodies detected in AIT are TPOAb and TgAb. TPOAb occurs in about 80–90% of patients with AIT, while TgAb is in approximately 40–70% of patients. In about 10% of patients with AIT, only TgAb is detected in the serum [6].

AIT is a multifactorial disease with different elements involved in its pathogenesis, including genetic (polymorphisms of (1) major histocompatibility genes; (2) immunoregulatory genes: CTLA4, PTPN22, FOXP3, CD25, CD40, FRCL3; (3) cytokines genes: IL1B, IL10 etc., and (4) thyroid-specific genes), environmental (iodine excess, vitamin D deficiency, selenium deficiency, stress, viral infections, drugs, other nutritional factors), and epigenetic (methylation pattern, non-coding RNAs, etc.) factors [7]. The combination of these elements leads to the breakdown of self-tolerance and the formation of autoreactive lymphocytes [8].

Although the pathogenesis of AIT is not yet fully understood, among many gene polymorphisms, there is a postulated role of thyroid-specific genes, i.e., thyroid peroxidase (*TPO*) and thyroglobulin (*Tg*). The direct TPOAb role in the pathogenesis of AIT is questioned and some authors regard it rather as a hallmark of the disease [9]. Contrarily, it cannot be excluded that changes in the structure and expression of the protein or genetically determined interaction between antigen and antigen-presenting HLA molecule could be one of the steps on the pathogenetic path [10]. TPOAb concentration was shown to correlate with symptoms and quality of life of AIT patients regardless of thyroid function [11]. There are some studies showing an association between selected single nucleotide polymorphisms (SNPs) of the *TPO* gene and the risk of AIT [12,13,14,15]. However, the number of studies to date is limited. The aim of our study was to assess the frequency of selected *TPO* gene promotor SNPs: rs2071399 G/A, rs2071400C/T, rs2071402 A/G, and rs2071403 A/G in AIT patients compared to controls in the Polish population.

## 2. Results

There were no significant differences in the genotype or allele frequencies of the studied TPO gene promoter polymorphisms between patients and the control group. The observed allele and genotype frequencies were similar in patients and controls. No significant differences were found between the groups, even after taking into account different models of inheritance (dominant or recessive). Detailed results are presented in Table 1.

Haplotype analysis was performed. According to D’ values, all analyzed SNPs were in strong LD. R^2^ values showed that the strongest LD was observed between rs2071399 and rs2071403 (r^2^ = 0.87) or between rs2071402 and rs2071403 (r^2^ = 0.72). The weakest observed LD was found between rs2071400 and rs2071402 (r^2^ = 0.05). A haplotype block of three SNPs (rs2071400C/T, rs2071402 A/G, and rs2071403 A/G) was established, according to Gabriel et al. [16]. In both the patient group and the control group, the most frequent haplotype was CGA (48.1% and 51.2%, respectively); the differences between the groups were not significant. The second most frequent haplotype was CAG (42.3% in AIT and 43.0% in the control group), with no significant differences between the analyzed groups. A rare haplotype, CGG, was absent in controls and occurred in 1.7% of AIT patients (*p* < 0.05). Haplotype frequencies and p values are shown in Table 2.

The association between *TPO* gene SNPs and some ultrasonographic features was further analyzed in a subgroup of 116 AIT patients. They were divided into (1) AIT with nodules and (2) AIT without nodules. However, it was shown that these subgroups did not differ significantly in the distribution of genotypes of the studied SNPs (*p* > 0.05 in the case of all four studied SNPs). We also checked for the association between genotypes of the studied *TPO* gene SNPs and thyroid volume. Unfortunately, it was shown that there were no significant differences in thyroid volume between different genotypes (*p* > 0.05 in the case of all studied SNPs).

## 3. Discussion

Hashimoto’s thyroiditis is the most common organ-specific autoimmune disease in humans [17]. It is the most prevalent cause of hypothyroidism [13]. There are observations of lower quality of life and persistent symptoms in the course of AIT, independent of hypothyroidism [18]. Therefore, it would be of great benefit to identify the population at high risk of developing AIT and implement a preventive approach (although there are still no proven methods available) [19]. Thyroid-specific genes are seen as interesting candidates, including primarily *TPO* and *Tg* genes in the case of AIT and *TSHR* genes in Graves’ disease (GD). These genes encode the main thyroid autoantigens in the course of autoimmune thyroid diseases (AITD). The *Tg* gene was shown to be associated with a predisposition to AIT in various populations [20,21,22,23,24,25,26,27]. It is postulated that some *Tg* SNPs modify protein structure and, thus, change the immunogenicity of the molecule [20]. Alternatively, different promoter region SNPs influence the affinity of *Tg* to bind transcription factors [28]. Similarly, some polymorphisms of the *TSHR* gene were shown to cause increased susceptibility to GD. Alterations in the TSH receptor molecule may probably affect both central and peripheral immune tolerance mechanisms [29].

*TPO* encodes a thyroid-specific 110 kDa glycosylated hemoprotein bound to the apical membrane of the thyrocyte. The 933 amino acid-containing protein is synthesized from TPO mRNA of 3 Kb length [30]. The *TPO* gene contains 17 exons, its size is 150 Kb, and it is located on chromosome 2 at the 2p25 locus [31,32]. The lack of TPO activity implies the inability to iodine tyrosine residues in Tg and to couple these residues into thyroid hormones, mainly T4 and, in smaller proportions, T3 and rT3 [33,34]. TPO mutations are among the main causes of thyroid dyshormonogenesis, resulting in congenital hypothyroidism and goiter [35]. Moreover, TPO is one of the main autoantigens (along with Tg) being a target of the immune system in the course of AIT. TPOAb antibodies can be detected in approximately 95% of AIT patients [7].

To date, several different SNPs of the TPO gene have been examined for their association with AIT risk. Among them, four different promotor region SNPs were analyzed in this paper. Table 3 summarizes the results of studies on these four TPO promoter polymorphisms. Tomari et al. confirmed that in the case of two promoter SNPs, rs2071400 C/T and rs2071403 A/G, there were significant differences in allele and genotype frequencies between patients (all AITD and AIT group) and controls in the Japanese population. Moreover, there was also an association between rs2071400 C/T SNP and the level of TPOAb antibodies [13]. Two other TPO promoter SNPs (rs2071399 G/A and rs2071402 A/G) were distributed similarly in AITD patients and controls in the Japanese study [13]. The potential role of the rs2071400 C/T SNP in the development of AIT was also confirmed in the Egyptian cohort (no other promoter region SNPs were studied) [15]. Rs2071403 TPO SNP was also earlier shown to be associated with TPOAb antibody positivity (but not with hypothyroidism) in the Korean genome-wide association study (GWAS) [36]. Our study was the first attempt to examine the association between TPO promoter SNPs and AIT in the Caucasian population. As presented above, we did not replicate the observations of Tomari et al. and Ahmed et al. It cannot be ruled out that the analyzed SNPs are ethnically specific, but at the same time, different SNPs of the TPO gene may play a role in the European Caucasian population. The observed differences in allele frequencies of the studied SNPs in Europe and Asia (including Japan) may also be a partial explanation for the differences in study results (rs2071399, frequency in Europe A = 0.52; G = 0.48 vs. Japan A = 0.36; G = 0.64, https://www.ncbi.nlm.nih.gov/snp/rs2071399 accessed on 3 February 2024; rs2071400, frequency in Europe C = 0.92; T = 0.08 vs. Japan C = 0.65; T = 0.35, https://www.ncbi.nlm.nih.gov/snp/rs2071400 accessed on 3 February 2024; rs2071402 frequency in Europe G = 0.58; A = 0.42 vs. Japan G = 0.82; A = 0.18, https://www.ncbi.nlm.nih.gov/snp/rs2071402 accessed on 3 February 2024; rs2071403, frequency in Europe A = 0.50; G = 0.50 vs. Japan A = 0.37; G = 0.63, https://www.ncbi.nlm.nih.gov/snp/rs2071403 accessed on 3 February 2024). A possible limitation of our study is potentially the size of the studied group, which may be too small to confirm some weak associations. It would be best to assess these TPO gene SNPs in a larger European cohort. Another TPO promoter region SNP—rs11211645 G/A—was found to be significantly associated with the risk of AIT in Polish patients, although its effect was assessed as weak [14].

The mechanism of the influence of SNPs in the *TPO* gene promoter region on the development of the disease is not fully explained. It is expected that SNPs located in this region may affect protein expression due to their location close to transcription factor binding sites and, as a result, potential changes in the binding affinity of transcription factors. In the case of the rs2071403 A/G SNP, it was shown that a change in allele can modify mRNA expression in thyrocytes [36]. There is no direct evidence for the mechanism of action of other SNPs of the *TPO* promoter region.

Other *TPO* SNPs located outside the gene promoter region were also studied in AIT patients, although the number of studies is limited. In a paper from 2016, Brcic et al. tested one *TPO* SNP, rs11675434 C/T, among different SNPs located within genes other than *TPO*. The authors showed that this intronic variant is associated with AIT development risk and TPOAb antibody levels in the Croatian Caucasian population [12]. Rs11675434 was also shown to be associated with TPOAb positivity in two large Chinese studies [37,38], with TPOAb level in a Swedish paper [39], and with both TPOAb positivity and its level in a meta-analysis of different Caucasian GWAS studies (including cohorts from USA, Australia and multiple European countries) [40]. Ahmed et al. evaluated three other *TPO* SNPs of different locations within the gene (rs732609 A/C, rs1126797 C/T, and rs2071400 C/T) in Egyptians diagnosed with AIT. Only rs732609 A/C *TPO* SNP was shown to be associated with the disease risk [15]. In a previously mentioned study by Jabrocka-Hybel et al., two other SNP outside the promoter region were also studied (rs961028 and rs2276704), although they did not show any significant association with the risk of AIT [14]. Tomari et al. studied in a previously mentioned paper also some different *TPO* SNPs outside the promoter region (rs1126799 C/T, rs1126797 T/C, rs732609 A/C, and rs2048722 A/G), but their role in AIT development has not been confirmed. Rs2048722 A/G showed only some association with TPOAb positivity [13]. Some associations between *TPO* SNPs and TPOAb levels were also found in Iranian patients [41,42].

*TPO* gene SNPs were also evaluated for their role in Graves orbitopathy (GO) pathogenesis. In a study of Caucasian Polish patients with GO, rs11675434 *TPO* SNP was found to influence the development and course of the disease [43]. However, this SNP showed no association with the risk of GD [44].

To date, no single strong genetic factor has been found to be responsible for AIT development. Even in the case of proven factors, odds ratio (OR) values. There are also large ethnic and geographic differences. Nevertheless, further searches for genetic factors co-responsible for the development of AIT seem to be advisable. The development of molecular biology methods and the decline in the prices of genetic tests, which allow for the testing of many genes at the same time, should, in the future, allow the creation of screening panels determining the risk of developing a given disease.

## 4. Materials and Methods

### 4.1. Patients

The study group consisted of 237 unrelated patients diagnosed with AIT. The diagnosis of AIT was based on standard diagnostic criteria as previously described [45]. At least three of four mentioned (1) clinical symptoms of hypothyroidism, (2) hypo- or euthyroidism found by thyroid function tests (TSH), (3) elevated anti-thyroid antibodies (thyroid peroxidase antibodies—TPOAb and/or thyroglobulin antibodies—TgAb), and (4) thyroid ultrasound features typical for AIT. In our study, all patients included were hypothyroid (overt or subclinical hypothyroidism) and required levothyroxine supplementation. The control group consisted of 130 healthy individuals with no personal or family history of thyroid dysfunction (TSH within normal range), goiter, AITD, or other autoimmune disorders. Mean age in AIT group was 48.71 ± 13.85 (ranging from 18 to 79) compared to 33.61 ± 14.42 in the control group (18–74). Female-to-male ratio was 24.13:1 in AIT vs. 3.6:1 in the control group. Both patients and controls were of Caucasian Polish origin and came from the same geographic area. Informed consent to participate in the study was obtained from each patient in the AIT and control groups. This study was approved by the local ethics committee.

Detailed and comparable thyroid ultrasound reports were available for a group of 116 AIT patients. According to the presence or absence of thyroid nodules (with nodules—at least one solid lesion ≥5 mm in diameter found; without nodules—no solid lesions ≥5 mm in diameter), these patients were divided into two groups.

### 4.2. Methods

Four SNPs located in the promoter region of *TPO* gene were studied (rs2071399 G/A, rs2071400 C/T, rs2071402 A/G, rs2071403 A/G). Genomic DNA was isolated from whole blood using NucleoSpin Blood silica spin columns (Macherey-Nagel, Düren, Germany) according to the manufacturer’s instructions. A fragment of the *TPO* gene with a size of 1070 bp was amplified by PCR using the Color Perpetual Opti Taq PCR kit E2775 (EURx Ltd., Gdańsk, Poland) with the following program: (1) predenaturation, 5 min at 94 °C; (2) amplification, 35×; denaturation, 30 s at 94 °C, annealing, 30 s at 58 °C; elongation, 1 min at 72 °C; (3) final extension, 7 min at 72 °C. The primer sequences were as follows: 5′-CTGAAGCCTTTGCATCGTGT-3′, 5′-CGCCACCCATACCTGCTATA-3′. Amplicons underwent column-based clean-up, NanoDrop concentration measurement and were Sanger sequenced from both sides at the Molecular Biology Techniques Laboratory of Adam Mickiewicz University, Poznan, Poland. All four SNPs were determined using ChromasPro v.2.1.10 (Technelysium Pty Ltd, South Brisbane, Australia) and re-checked manually (see Figure 1).

### 4.3. Statistical Analysis

In both patients and controls, the distribution of genotypes was tested for deviations from Hardy–Weinberg equilibrium (chi-squared test—χ2) [46]. Allele and genotype frequencies in the AIT group and controls (or AIT with and without nodules) were compared with the use of χ2, Fisher’s exact, or χ2 for trend tests, where appropriate. Odds ratios (OR) with 95% confidence intervals (95% CI) were calculated as a measure of the strength of associations. To compare thyroid volume between different genotype subgroups, Kruskal–Wallis test was used. GraphPad Prism version 9.5.1 for Windows (GraphPad Software, Boston, MA, USA) was used for calculations. A *p*-value of less than 0.05 was considered statistically significant. Linkage disequilibrium (LD) between the studied polymorphisms and haplotype frequencies was calculated with further comparison between patients and controls (Haploview 4.2 version, Broad Institute, Cambridge, MA, USA) [47].

## 5. Conclusions

Contrary to some previous results, there was no association of the *TPO* gene promoter SNP with susceptibility to AIT in the Caucasian Polish population. Further research into different populations is needed to resolve this discrepancy. Furthermore, polymorphisms of different regions of *TPO* should be assessed to elucidate the role of this thyroid-specific gene in the pathogenesis of AIT.

## Figures and Tables

**Figure 1 ijms-25-03312-f001:**
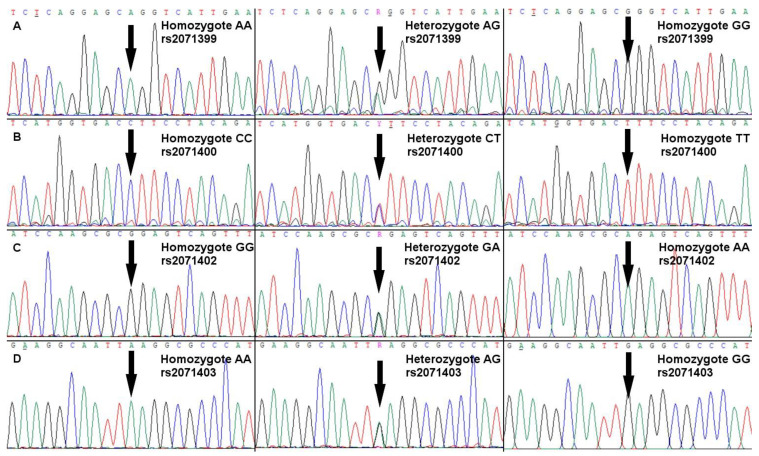
DNA sequencing results of the studied SNPs of the *TPO* gene promoter. The arrows indicate polymorphisms’ position. (**A**) rs2071399, location: chr 2: 1412756; A > C/A > G/A > T; (**B**) rs2071400, location chr2: 1412867; C > T; (**C**) rs2071402, location chr2:1413427; G > A/G > T; (**D**) rs2071403, location chr2:1413472, A > C/A > G/A > T.

**Table 1 ijms-25-03312-t001:** Frequency of allele and genotypes of selected SNPs located in the promoter of the *TPO* gene in patients with AIT and in the control group in the Polish population.

SNP	Genotype/Allele	AIT (%)	Controls (%)	*p* Value	OR (95% CI)
rs2071399	AA	51 (21.52)	37 (28.46)	0.21 * 0.52 ***	
	AG	123 (51.90)	56 (43.08)		
	GG	63 (26.58)	37 (28.46)		
	GG vs. AA + AG	63 (26.58) vs. 174 (73.42)	37 (28.46) vs. 93 (71.54)	0.70 *	0.91 (0.56–1.47)
	AG + GG vs. AA	186 (78.48) vs. 51 (21.52)	93 (71.54) vs. 37 (28.460	0.14 *	1.45 (0.89–2.37)
	A allele	225 (47.47)	130 (50.00)	0.51 *	1.11 (0.82–1.50)
	G allele	249 (52.53)	130 (50.00)		
rs2071400	CC	203 (85.65)	116 (89.23)	0.62 ** 0.29 ***	
	CT	33 (13.92)	14 (10.77)		
	TT	1 (0.42)	0 (0)		
	TT vs. CC + CT	1 (0.42) vs. 236 (99.58)	0 (0) vs. 130 (100)	1.0 **	1.66 (0.07–40.93)
	CT + TT vs. CC	34 (14.35) vs. 203 (85.65)	14 (10.77) vs. 116 (89.23)	0.33 *	1.39 (0.72–2.69)
	C allele	439 (92.62)	246 (94.62)	0.35 **	1.40 (0.74–2.65)
	T allele	35 (7.38)	14 (5.38)		
rs2071402	AA	42 (17.80)	26 (20.00)	0.86 * 0.78 ***	
	AG	118 (50.00)	62 (47.69)		
	GG	76 (32.20)	42 (32.31)		
	GG vs. AG + AA	76 (32.20) vs. 160 (67.80)	42 (32.31) vs. 88 (67.69)	0.98 *	1.00 (0.63–1.57)
	GG + AG vs. AA	194 (82.20) vs. 42 (17.80)	104 (80.00) vs. 26 (20.00)	0.60 *	1.15 (0.67–1.99)
	A allele	202 (42.80)	114 (43.85)	0.78 *	1.04 (0.77–1.42)
	G allele	270 (57.20)	146 (56.15)		
rs2071403	AA	56 (23.73)	35 (27.13)	0.75 * 0.47 ***	
	AG	118 (50.00)	63 (48.84)		
	GG	62 (26.27)	31 (24.03)		
	GG vs. AA + AG	62 (26.27) vs. 174 (73.73)	31 (24.03) vs. 98 (75.97)	0.64 *	1.13 (0.69–1.85)
	AG + GG vs. AA	180 (76.27) vs. 56 (23.73)	94 (72.87) vs. 35 (27.13)	0.47 *	1.20 (0.73–1.95)
	A allele	230 (48.73)	133 (51.55)	0.47 *	1.12 (0.83–1.52)
	G allele	242 (51.27)	125 (48.45)		

AIT—autoimmune thyroiditis; OR—odds ratio; CI—confidence interval. * Chi^2^ test, ** Fisher exact test, *** Chi^2^ test for trend.

**Table 2 ijms-25-03312-t002:** Haplotype frequency of three SNPs of the promoter region of the *TPO* gene (rs2071400C/T, rs2071402 A/G, and rs2071403 A/G) in patients with AIT and in the control group in the Polish population.

Haplotype	AIT (%)	Controls (%)	*p* Value
CGA	48.1	51.2	0.4269
CAG	42.3	43.0	0.8586
TGG	7.2	5.4	0.3472
CGG	1.7	0.0	**0.0356**

AIT—autoimmune thyroiditis.

**Table 3 ijms-25-03312-t003:** The role of four selected SNPs of the TPO gene promoter (rs2071399, rs2071400, rs2071402, rs2071403) in the pathogenesis of AIT.

Study	Population	Studied Group	Studied SNP	Results	Method
Tomari S. et al. Endocrine J, 2017 [13]	Japanese	AIT 147 Controls 92	rs2071399 G/A rs2071400 C/T rs2071402 A/G rs2071403 A/G	NS T allele (CT + TT) more frequent in AITD, including AIT; TT more frequent in AIT NS G allele and GG more frequent in AITD, including AIT	RFLP
Ahmed H.S. et al. Endocr Metab Immune Disord Drug Targets, 2021 [15]	Egyptian	AIT 200 Controls 100	rs2071400 C/T	T allele and TT genotype significantly more frequent in AIT	RFLP

AIT—autoimmune thyroiditis; NS—not significant; RFLP—restriction fragment length polymorphism.

## Data Availability

Data are contained within the article.

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
