# Peer review of "Is There a Link between Thyroid Peroxidase Gene Promoter Polymorphisms and Autoimmune Thyroiditis in the Polish Population?"

_ijms, 2024, doi:10.3390/ijms25063312_

Round 1
Reviewer 1 Report
Comments and Suggestions for Authors
ID: ijms-2887860
Is There A Link Between Thyroid Peroxidase Gene Promoter Polymorphisms And Autoimmune Thyroiditis In The Polish Population? by Lacka, et al.
To the Authors:
General comments:
The authors investigated the role of selected TPO gene promoter polymorphisms (rs2071399 G/A, rs2071400C/T, rs2071402 A/G, and rs2071403 A/G) in the development of autoimmune thyroiditis, which is for the first time in the Caucasian population. 237 patients diagnosed with autoimmune thyroiditis (AIT) and 130 healthy controls were included. They found that there were no significant differences in the genotype and allele frequencies of the studied TPO gene promoter polymorphisms between patients and controls. On the other hand, AIT patients had a rare haplotype (CGG) more frequently. It was considered that this study was structured well and included novelty; however, several points should be addressed to improve the manuscript.
Specific comments:
1. In the abstract, the authors described a difference of p<0.05 in the genotype and allele frequencies of the studied TPO gene promoter polymorphisms between patients and controls in line 24; however, the p-value is not found in the main text. Please clarify this point.
2. Were the TPO gene promoter polymorphisms associated with the severity and intractability of AIT? Were serum levels of TPO antibodies related to the TPO gene promoter polymorphisms in AIT patients?
3. Since the number of included patients was relatively small, the authors should mention this limitation in the manuscript.
4. How about the polymorphism of the thyroglobulin (Tg) promoter gene, since the specific antibody for the Hashimoto’s thyroiditis is anti-Tg antibody?
5. Were there any clinical relevance of the TPO gene polymorphism to the titer of anti-TPO antibody and/or any complications of the Hashimoto’s thyroiditis?
Author Response
- In the abstract, the authors described a difference of p<0.05 in the genotype and allele frequencies of the studied TPO gene promoter polymorphisms between patients and controls in line 24; however, the p-value is not found in the main text. Please clarify this point.
Thank you for your remark, it will be corrected in the manuscript (should be p>0.05). All differences in allele and genotype frequencies between patients and controls were nonsignificant, as described in the text and shown in table.
- Were the TPO gene promoter polymorphisms associated with the severity and intractability of AIT? Were serum levels of TPO antibodies related to the TPO gene promoter polymorphisms in AIT patients?
Detailed and comparable clinical data were available only for some part of the studied group. It makes subgroup analysis difficult to interpret because of small number of patients.
All patients were hypothyroid (subclinical or overt) and therefore on levothyroxine substitution.
We added the analysis of the association between studied SNPs and selected ultrasonographic features (thyroid volume, presence of thyroid nodules) in a subgroup of 116 AIT patients. As presented in the manuscript, we did not find any significant associations in that field.
Great majority of patients has elevated level of TPOAb and/ or TgAb. Unfortunately these parameters were measured in variable laboratories with different kits used. Moreover, it is known that antibodies level fluctuate over time and usually decrease with time since the diagnosis. As our group of AIT patients was not homogenous in this aspect (both newly diagnosed patients and those with long-term disease). All this makes the analysis of the relationship between antibody concentration and polymorphisms difficult to interpret
- Since the number of included patients was relatively small, the authors should mention this limitation in the manuscript.
Thank you for your suggestion, it has been added in the text. We agree that larger, international European cohorts could give more reliable answer about the role of these SNPs in AIT pathogenesis in Caucasian population.
- How about the polymorphism of the thyroglobulin (Tg) promoter gene, since the specific antibody for the Hashimoto’s thyroiditis is anti-Tg antibody?
It would be also a matter of our interest to analyze these SNPs in the future.
- Were there any clinical relevance of the TPO gene polymorphism to the titer of anti-TPO antibody and/or any complications of the Hashimoto’s thyroiditis?
We have analyzed the potential role of these SNPs for some ultrasonographic features of AIT – thyroid volume and thyroid nodule presence. However, we failed to find such relationship (this part was added to result section).
Since TPOAb level was not assessed by one method in our study (patients from ambulatory care units, parameters measured by random commercial studies) and patients were diagnosed at variable time points since the diagnosis of AIT we believe that no such assessment can be made and give objective response for such influence. TPOAb level can vary over time and usually decreases with time after initiation of autoimmune process.
Reviewer 2 Report
Comments and Suggestions for Authors
Thank you very much for having the opportunity to review this paper. In this manuscript, the authors evaluated the role of selected TPO gene promoter polymorphisms in the development of autoimmune thyroiditis. I believe that the authors have made an interesting contribution in their field with the current research.
I have written some suggestions to further improve the study. Below are my specific comments.
Abstract: According to the Instructions for authors, the abstract should be a single paragraph and should follow the style of structured abstracts, but without headings.
Line 16: Autoimmune thyroiditis (AIT) - the abbreviation must appear the first time, after which only this is used.
Results: Why was the haplotypes analysis performed only for three SNPs and rs2071399 G/A was excluded?
Materials and Methods:
Line 177: “standard diagnostic criteria as previously described” - these criteria should be briefly listed.
What was the age range and gender distribution of the groups?
For reagents and materials, the city and country of the manufacturers should also be specified.
In the legend of Figure 1, there should not appear data regarding SNPs frequencies and references.
The percentage of iThenticate report (45%) is rather high. The text should be revised in this regard.
Author Response
Abstract: According to the Instructions for authors, the abstract should be a single paragraph and should follow the style of structured abstracts, but without headings.
Thank you for your remark, it has been corrected in the text.
Line 16: Autoimmune thyroiditis (AIT) - the abbreviation must appear the first time, after which only this is used.
Thank you for your remark, it has been corrected in the text.
Results: Why was the haplotypes analysis performed only for three SNPs and rs2071399 G/A was excluded?
Haplotype blocks have been constructed according to Gabriel et al. method (confidence interval) by Haploview 4.2 software. This is a preferred method by some authors. If we would use different definition of haplotype blocks (solid spine of LD or Four Gamete Rule) all four SNPs would be included. There was no significant difference between patients and controls when comparing frequencies of haplotypes that included all four SNPs.
Materials and Methods:
Line 177: “standard diagnostic criteria as previously described” - these criteria should be briefly listed.
Thank you for your suggestion, it has been corrected in the text.
What was the age range and gender distribution of the groups?
Thank you for your remark, it has been added in the text.
For reagents and materials, the city and country of the manufacturers should also be specified.
Thank you for your suggestion, it has been corrected in the text.
In the legend of Figure 1, there should not appear data regarding SNPs frequencies and references.
It has been changed and moved to discussion section.
The percentage of iThenticate report (45%) is rather high. The text should be revised in this regard.
Some sentences have been rewritten. The authors have no iThenticate. Grading with Grammarly results in a lower repeat score.
Reviewer 3 Report
Comments and Suggestions for Authors
The primary objective of the present study was to examine the impact of specific TPO gene promoter polymorphisms (rs2071399 G/A, rs2071400C/T, rs2071402 A/G, and rs2071403 A/G) on the development of autoimmune thyroiditis (AIT). The authors have genotyped 237 AIT-diagnosed patients and 130 healthy controls for the four TPO gene polymorphisms. The results indicated no significant differences in genotype and allele frequencies of the examined TPO gene promoter polymorphisms between patients and controls (p < 0.05). The haplotype distribution (rs2071400-rs2071402-rs2071403) was comparable between the two groups for the most common variants (CGA, CAG, TGG). Notably, a rare haplotype (CGG) exhibited a higher occurrence among patients compared to controls (p=0.04). In conclusion, the investigated TPO gene promoter polymorphisms did not exhibit an association with susceptibility to AIT in the Caucasian Polish population, contrary to previously published findings.
The study is interesting and well-performed. The statistical analysis is appropriate. The authors have made the effort to discuss previous studies.
Suggestions for improvement
The authors should expand the rationale for analyzing TPO gene polimorphisms in the Introduction section.
Also, in the Discussion section, they should better focus on the future directions of their study concerning TPO (and/or other genes) polymorphisms in AIT
Comments on the Quality of English LanguageMinor editing required
Author Response
The authors should expand the rationale for analyzing TPO gene polimorphisms in the Introduction section.
This part has been rewritten and expanded.
Also, in the Discussion section, they should better focus on the future directions of their study concerning TPO (and/or other genes) polymorphisms in AIT
This part has been rewritten and slightly expanded.
Round 2
Reviewer 1 Report
Comments and Suggestions for Authors
ID: ijms-2887860
Is There A Link Between Thyroid Peroxidase Gene Promoter Polymorphisms And Autoimmune Thyroiditis In The Polish Population? by Lacka, et al.
To the Authors:
General comments:
The authors appropriately revised the manuscript according to the comments.